# Methylation in the Promoter Region of the Dopamine Transporter *DAT1* Gene in People Addicted to Nicotine

**DOI:** 10.3390/ijerph19148602

**Published:** 2022-07-14

**Authors:** Jolanta Chmielowiec, Krzysztof Chmielowiec, Aleksandra Strońska-Pluta, Aleksandra Suchanecka, Kinga Humińska-Lisowska, Milena Lachowicz, Marta Niewczas, Monika Białecka, Małgorzata Śmiarowska, Anna Grzywacz

**Affiliations:** 1Department of Hygiene and Epidemiology, Collegium Medicum, University of Zielona Góra, 65-046 Zielona Gora, Poland; chmiele1@o2.pl (J.C.); chmiele@vp.pl (K.C.); 2Independent Laboratory of Health Promotion, Pomeranian Medical University in Szczecin, 70-204 Szczecin, Poland; aleksandra.stronska@pum.edu.pl (A.S.-P.); o.suchanecka@gmail.com (A.S.); 3Faculty of Physical Education, Gdansk University of Physical Education and Sport, 80-336 Gdansk, Poland; kinga.huminska-lisowska@awf.gda.pl; 4Department of Psychology, Gdansk University of Physical Education and Sport, 80-336 Gdansk, Poland; millkawings@gmail.com; 5Faculty of Physical Education, University of Rzeszow, 35-959 Rzeszow, Poland; mniewczas@ur.edu.pl; 6Department of Pharmacokinetics and Therapeutic Drug Monitoring, Pomeranian Medical University, 70-111 Szczecin, Poland; monika-bialecka@post.pl (M.B.); malgorzata.smiarowska@wp.pl (M.Ś.)

**Keywords:** *DAT1* gene, promoter methylation, nicotine addiction

## Abstract

The dopaminergic system is a crucial element of the addiction processes. The dopamine transporter modulates the dynamics and levels of released dopamine in the synaptic cleft. Therefore, regulation of dopamine transporter (*DAT1*) gene expression is critical for maintaining homeostasis in the dopaminergic system. The aim of our study is evaluation of the methylation status of 33 CpG islands located in the *DAT1* gene promoter region related to nicotine dependency. We investigated 142 nicotine-dependent subjects and 238 controls. Our results show that as many as 14 of the 33 CpG islands tested had statistically significantly higher methylation in the nicotine-dependent group compared to the control group. After applying Bonferroni correction, the total number of methylation sites was also significantly higher in the dependent subjects group. The analysis of the methylation status of particular CpG sites revealed a new direction of research regarding the biological aspects of nicotine addiction.

## 1. Introduction

Cigarette smoking is one of the leading causes of preventable death in the United States and other developed countries. According to the US Centers for Disease Control and Prevention, more than 480,000 deaths annually are caused by cigarettes, including deaths from secondhand smoke. The life expectancy for smokers is at least ten years shorter than for nonsmokers [1,2]. In the United States, more than 5 million Americans under 18 years old are projected to die from a smoking-related disease. Worldwide, about one billion people aged 15 years and above [3] and an estimated twenty-four million children aged 13–15 years smoke [4].

Epigenetics is concerned with the processes that modify the regulation of genes without altering the DNA nucleotide sequence. These processes may result in phenotype differentiation [5,6,7,8,9]. Consequently, genes can be turned on or off, which is also associated with exposure to certain factors, such as psychoactive substances. Epigenetics may become an important aspect of clinical diagnostics. 

This includes the regulation of gene expression mediated by DNA methylation, nucleosome structure and position, post-translational modification of histones, replacement of histones, and RNA interference [10]. DNA methylation is the best known and most studied epigenetic mechanism. It consists of adding a methyl group to the C5 position of the cytosine at the CpG site [7,11] by DNA methyltransferase to form 5-methylcytosine (5-mC), which is an epigenetic marker that enables the expression or inhibition of genes [12] and may be passed on to the next generation. 

Methylation makes chromatin more condensed, i.e., less accessible to transcription factors. Studies have shown that DNA methylation is associated with multiple substance addictions [11,13]. Epigenetic changes, including methylation status, reflect a person’s environmental conditions and lifestyle. Therefore, epigenetic changes can be used as biological markers showing metabolic dysfunctions [8,14,15].

Moreover, DNA methylation governs gene expression by recruiting proteins that participate in gene repression or by inhibiting the binding of transcription factors to DNA. Among the distinguishing features of transcription factors is their possession of DNA-binding domains that enable them to bind to specific DNA sequences, called enhancer or promoter sequences. Certain transcription factors bind to the promoter sequence of DNA near the transcription start site, thus, assisting in forming the transcription initiation complex. Other transcription factors bind to regulatory sequences, such as enhancer sequences, and can either stimulate or suppress transcription of the associated gene. These regulatory sequences can be situated in thousands of base pairs upstream or downstream of the transcribed gene. Transcriptional regulation is the most prevalent form of gene regulation [16].

The regulation of DNA methylation has been analyzed in many physiological and behavioral phenotypes in animal models. In particular, it has been proven that it is involved in the development of neurons and the brain [5,17,18]. In human studies, methylation dysregulation has been observed in people with substance addiction, anxiety, depression, autism, schizophrenia, and bipolar disorder [13,18,19,20].

Up to decades after smoking cessation, it is associated with a long-term risk of disease, including certain cancers, chronic obstructive pulmonary disease, and stroke. The mechanisms behind these long-term effects are not well understood. Changes in DNA methylation have been proposed as one of the possible explanations [21,22]. The *DAT1* gene, encoding the human dopamine transporter, has been extensively studied under experimental and clinical conditions and is associated with various brain diseases and behavioral characteristics [23].

Dopamine (DA) neurotransmission is responsible for basic brain functions, such as movement, behavior, cognition, and motivation. Disruptions in dopamine signaling result in various disorders and neuropsychiatric states [24]. The dopamine transporter DAT plays a crucial role in DA signaling. It modulates the dynamics and DA levels in the synaptic cleft by recycling extracellular dopamine to the presynaptic terminal. Changes influence the concentration of synaptic dopamine and its reuptake kinetics and the availability of the dopamine transporter [25].

Substance dependence is a chronic, relapsing condition characterized by compulsive drug seeking [26,27]. The dopaminergic system plays an important role in the reinforcing effects of drug abuse [28,29]. The following three agents are involved in the development of addiction: genetic, diverse environmental factors, and the effects of drugs on gene expression or mRNA levels [30]. 

A recent study has shown that transcription factors, such as non-coding RNAs, histone modifications and chromatin structure, could change the transcriptional potential of genes. These transcription factors also contribute significantly to many neuroadaptations resulting from chronic drug exposure [29]. Increasing evidence supports the hypothesis that every mechanism of epigenetic regulation is directly affected by drugs of abuse. These adaptations are among the main processes through which drugs induce highly stable modifications in the brain that mediate the addicted phenotype [31].

Still, little is known about methylation in specific promoter regions among nicotine-dependent (ND) subjects. This study investigates the influence on nicotine addiction by asking whether the differences between smoking and not smoking affect the methylation of selected groups of promoters in the *DAT* gene. The aim of the study was to determine the methylation level in nicotine smokers. The aim was to check the biological dependence of the influence of the nicotine used on the metification level in the tested grapevine. We understand that there can be a two-way causal relationship between smoking and methylation; therefore, we analyzed this issue.

## 2. Materials and Methods

### 2.1. Participants

The study group comprised of 142 cigarette smoking subjects (mean age = 27.57, SD = 10.1; mean Fagerstrom test = 3.73, SD = 2.70; mean number of cigarettes consumed per day = 13.55, SD = 4.37), whereas the control group included non-smoking volunteers matched for age (mean age = 21.86 years, SD =3.55). The age of the participants in both groups is presented in Table 1. The exclusion criteria were an addiction to substances different than nicotine. Both groups consisted of people of European origin from the same region of Poland. All participants were European to reduce the possibility of genetic admixture and overcome any potential problems due to population stratification.

The study was approved by the bioethical committee of the Pomeranian Medical University in Szczecin and was conducted following the Declaration of Helsinki guidelines. Patients and the control group were informed about the principles of the study, familiarized with its course, and informed about the possibility of withdrawing from the study without giving any reason at any time during its duration. 

None of the participants in the study were financially rewarded for participating in the project, and the trials were anonymized entirely, in line with the principles of personal data protection. Nicotine addiction was tested with the Fagerstrom test. At the same time, the control group was selected according to age and sex among people who did not smoke more than a pack of cigarettes and were not addicted to nicotine or other psychoactive substances at the time of the study. All the procedures for allowing comfort and concentration were accomplished.

### 2.2. Methylation Status Assessment of Dopamine Gene Transporter (DAT1) Promoter

DNA was isolated from peripheral blood with a DNA isolation kit (A&A Biotechnology, Gdynia, Poland) as previously described [32]. After isolation, it was stored at −20 °C. Bisulfite modifications with 250 ng DNA were performed according to the manufacturer’s instructions using the EZ DNA Methylation Kit (Zymo Research, Orange, CA, USA) A methylation-specific PCR test was performed at Mastercycler epgradient S (Eppendorf, Germany).

Oligonucleotide primers designed using methprimer (http://www.urogene.org/cgi-bin/methprimer/methprimer.cgi, accessed on 29 April 2022) were obtained from Genomed.pl (Warsaw, Poland). The status of the DAT1 promoter (ENSG00000142319) was assessed by PCR using primers specific to a fragment of the gene, i.e., DATF: 5′-GGTTTTTGTTTTTTTTATTGTTGAG-3’; DATR: 5′-AAATCCCCTAAACCTAATCCC-3’. Table 2 shows the PCR conditions used to amplify the 447 bp fragment spanning the 33 CpG sites in the promoter of the *DAT1* gene.

The concentration of magnesium chloride ions was 2.5 mM. After the amplification assay, the PCR products were subjected to sequencing as previously described [32]. Briefly, the samples were verified by way of sequencing using the BigDye v3.1 kit (Applied Biosystems, Darmstadt, Germany) and separation by ethanol extraction using the ABI Prism 3130XL (Applied Biosystems, Darmstadt, Germany) in a 36 cm capillary in a POP7 polymer, using the reverse primer.

Sequencing chromatograms were analyzed using 4peaks software (Mek & Tosj, Amsterdam, The Netherlands). Methylation of cytosine was considered positive when the G/A+G ratio accounted for at least 20% of a total signal. The mathematical equation to calculate the percentage of methylation in each participant was (G/(G + A) × 100) [33].

### 2.3. Statistical Analysis

Normality of the distribution was not met for the analyzed variables. Analysis and comparison of the total methylation sites and age were performed using the Mann–Whitney U test. The data were analyzed using the chi-squared test, with *p* < 0.05 regarded as statistically significant. The Bonferroni, multiple comparisons correction, was applied for these variables, and the accepted level of significance was 0.0015 (0.05/33). Pearson’s correlations were analyzed between the total methylation sites and number of cigarettes smoked in a group of nicotine-dependent subjects (STATISTICA 13, TIBCO Software, Inc., Palo Alto, CA, USA; PQStat Software, v. 1.8.2., Poznań, Poland).

The mean total methylation sites (M_T_) was calculated using the following formula:$$M_{T}\;=\left(\left(n1\left(\Sigma ;\;\mathrm{MS}\;n1\right)+n2\left(\Sigma \;\mathrm{MS}n2\right)\ldots +nm\left(\Sigma \;\mathrm{MS}nm\right)\right)/n\right)$$

The mean percent total methylation sites (M_T%)_ was calculated using the following formula:$$MT_{\%}\;=\left(\;\frac{n1\left(\frac{\Sigma \;\mathrm{MS}\;n1}{33}\right)+n2\left(\frac{\Sigma \;\mathrm{MS}n2}{33}\right)\ldots +nm\left(\frac{\Sigma \;\mathrm{MS}nm}{33}\right)}{n}\right)\times 100\%$$
where

*n*1, *n*2, …, *nm*—the individual tested person;

Σ MS*n*1—the sum of the amount of sites methylation for a particular test person; and

*n*—the number of surveyed people.

## 3. Results

The analysis of the methylation status of particular CpG sites revealed differences in the methylation levels at particular sites (islands) in the *DAT1* promoter (Table 3 and Figure 1). At sixteen of the 33 CpG sites, a significantly higher methylation level was found in the nicotine-dependent group compared to the control (sites 1, 2, 4, 5, 7, 8, 9, 10, 11, 12, 13, 17, 20, 23, 26, and 27). Analysis of the total *DAT1* methylation revealed a statistically significant increase in the number of methylated islands in the nicotine-dependent group compared to the controls (M_T_ 20.819 vs. 14.916 (M_T%_ 63.09% vs. 45.20%); Z = 7.881, *p* = 0.000001).

Between the total methylation sites (M_T%_) of 33 CpG DAT1 sites and the number of cigarettes smoked in a group of nicotine-dependent subjects, the Pearson’s Correlation was positive (r = 0.273, *p* = 0.001, Figure 2).

## 4. Discussion

The dopamine transporter plays an important role in the neurotransmission of dopamine. It is located on the nerve endings and modulates the dynamics and levels of released dopamine by returning extracellular dopamine to the presynaptic terminal, thereby, terminating its function. Impaired dopamine activity may result from altered release or reuptake. For this reason, the regulation of *DAT1* gene expression is significant for maintaining homeostasis in the dopaminergic system [34].

In the present study, we analyzed 33 CpG sites located in the *DAT1* gene promoter region in nicotine-dependent and control subjects. Our analyses showed significant differences between the two groups—methylation changes were not similar in all sites. Compared to the control group, some sites in the dependent subjects were hypermethylated, and some were hypomethylated. Furthermore, an assessment of the ability of transcription factors to bind the indicated sites revealed a significant number of these regulators of gene expression.

Our results show that as many as 14 of the 33 CpG islands tested had statistically significantly higher methylation in the test group compared to the control group. As can be observed in Table 2, even if there was no statistical significance, in the vast majority of the CpG islands tested, the methylation level was significantly higher in the test group. In the case of islands 19, 22, 32, and 33, lower methylation levels were observed in the control group compared to the test group; however, these results are not statistically significant. Both studied groups—smokers and nonsmokers—were matched in terms of age and gender and came from the same population (common environmental factors)—the main differentiating factor was cigarette smoking, which indicates the influence of this factor on the increase in methylation in the promoter region of the *DAT1* gene.

Cigarette smoking continues to be common despite well-publicized adverse health effects [35]. It is well known that active smoking is a significant risk factor for cancer, cardiovascular disease and chronic obstructive pulmonary disease [36,37,38]. Prenatal exposure to cigarette smoke reduces fetal growth in the prenatal period, increases the risk of sudden infant death syndrome after birth, and promotes the development of addictive behaviors, immune system abnormalities, obesity, and associated cardiometabolic diseases after birth [39,40,41,42]. Some of these effects may be due to cigarette smoke’s modulation of DNA methylation.

Cigarette smoke is considered one of the most potent environmental DNA methylation modifiers [43]. First, cigarette smoke can modulate it by damaging DNA and then recruiting DNMT. Cigarette smoke carcinogens, such as arsenic, chromium, formaldehyde, polycyclic aromatic hydrocarbons, and nitrosamines [44,45], can damage DNA, causing double-strand breaks, as shown in the mouse embryonic stem of exposed cells to cigarette smoke condensate. In these experiments, surviving cells show a high ability to repair DNA and normal karyotypes [46]. DNA repair sites recruit DNMT1 [47], which methylates the CpG adjacent to the repaired nucleotides [48]. 

Second, cigarette smoke can also modulate DNA methylation through the effect of nicotine on gene expression [49]. Third, cigarette smoke can indirectly alter DNA methylation by modulating the expression and activity of DNA binding agents. For example, cigarette smoke condensate increased Sp1 expression and DNA binding in lung epithelial cells [50,51]. Sp1 is a common transcription factor that binds to GC-rich motifs in gene promoters [52] and plays a key role in early development; as such, it may prevent CpG de novo methylation within these motifs during early embryogenesis [53]. 

Fourth, cigarette smoke can alter DNA methylation through hypoxia—cigarette smoke contains carbon monoxide, which binds to hemoglobin (competitively with oxygen) and thus reduces tissue oxygenation [54]. Hypoxia leads to HIF-1α-dependent upregulation of methionine adenosyltransferase 2A, an enzyme that synthesizes S-adenosylmethionine, the leading biological methyl donor, which is critical to DNA methylation processes [55].

In a recent study by our team [56], we analyzed the same 33 CpG islands in a group of cannabis-dependent subjects. Three of the analyzed sites were possible PAX5 transcription factor binding sites (positions 3, 22, and 33), and all of them were hypomethylated in the group of dependent subjects. This is a crucial finding since PAX5 is a transcription factor associated with numerous processes, including the nervous system’s development [32]. Additionally, the transcription’s binding ability revealed that the CpG island covers a sequence that may be bound by ligand-bound GR (glucocorticoid receptor) (position 6). Interestingly, none of these islands differed significantly between the cases and controls in the present study. 

This may be vital since glucocorticoids mediate nervous system functions in substance dependency [35]. We concluded that hindering glucocorticoid responses could change drug-induced reactions, including drug-related learning and memory modulation. Additionally, we postulated that GR might become a promising target in substance use disorders and dependency therapy [57,58]. The DNA methylation level in the total CpG islands for DAT1 was higher in individuals without depression, anxiety, or ADHD family history compared to individuals with the above family histories (*p* < 0.05) [59]. 

The high scores of children’s ADHD problems were associated with high levels of DAT1 methylation at the M1 CpG site and with low levels of DAT1 methylation at the M2 and M6 CpG sites [60]. Recent studies confirmed how the interaction between genetic and environmental factors, and their impact on child and adolescent emotions and behaviors, might be mediated by epigenetic mechanisms of DNA methylation [61]. 

Traditional studies in epigenetic research underline the importance of the complex interaction between the *DAT1* gene and the environment [62]. The innovative study by De Nardi et al. [34] on *DAT1* 5′-UTR based on cross-correlations showed that specific patterns exist in the dynamics of CpG methylation. The results of studies done by the same method [63,64] showed that patients with Parkinson’s disease were characterized by an overall hyper-methylated condition in the 5′-UTR, the opposite from what was found in ADHD adolescents showed hypomethylated conditions [65,66]. Cross-correlations shown by Tafani et al. [64] considered evidence of both a methylated and a demethylated loci status. 

Significant relationships showed specific dynamics among the CpGs within and between the two motifs among the matched loci. The approach used in these studies investigated simple pairwise correlations between couples of individual CpGs methylated simultaneously, focusing on adolescents and young adult subjects. At this age, hyperactivation derives from neurobiological plasticity, which allows adolescents to adjust to the body and emotional transformations they undergo, including physical changes, emotional experiences, separation from internalized parental figures, and the construction of a personal, social, and sexual identity [67,68,69]. Therefore, youths in this phase could be at higher risk for psychopathology (including depression, anxiety, problematic use of the web, and internet addiction).

The epigenetic sensitivity of the *DAT1* gene increased during the process of evolution. The genetic drift of the *DAT1* sequence oriented on the accumulation of GC nucleotides may reflect its strengthening epigenetic potential, significant in the regulatory processes resulting from more complex functions of the human brain [70]

Peripheral tissues have significant limitations concerning generalizability to other tissues of interest, such as the brain. However, a growing amount of evidence proves that the numerous epigenetic changes found in peripheral leukocytes and transformed lymphoblasts also correspond to alterations in the brain cells [71]. Most importantly, a study performed by Wiers et al. demonstrated that peripheral *DAT1* promoter methylation might be a predictive factor of dopamine transporter availability in the striatum [62].

It is also known that addictions are not one-dimensional and that their occurrence requires unknown accompanying factors. Although different mechanisms regulate epigenetics and genetics, the effects of their disorders may be cumulative. Thus, if individual genetic variants have a risk effect of a few percent, perhaps epigenetic factors may increase or decrease that risk. The role of environmental factors in these mechanisms is prominent. 

However, due to their close correlation with epigenetics, it seems that the simultaneous study of measurable factors, such as SNPs and the methylation level of gene promoters with proven importance in pathogenesis, is a key aspect in the search for the pathomechanisms of addiction. The attempt to identify the mechanisms of the addiction process in relation to two different mechanisms is an innovative approach to the problem; the secret of the biological basis of addiction is the subject of many intensive studies.

As we can see, it is justified to study epigenetic changes, especially the methylation of promoter sites of selected genes in research on addiction. It should be emphasized that multiple genes and factors that characterize addictions are of great importance and are also an obstacle. Our team’s research focuses on genetic factors while ignoring many analyses, and large groups of patients with diagnosed addiction were analyzed; however, we still do not know a specific genetic pattern as there are other factors, such as psychological and environmental factors and those related to past traumas or stress. However, what we emphasize and what we have proven in our research justifies a focus on islet methylation in the regions of promoters associated with candidate genes in addiction. This is a crucial aspect of understanding the pathology of addiction, especially from the biological point of view.

## 5. Conclusions

Our analysis of the methylation status of particular CpG sites revealed a new research direction regarding the biological aspects of nicotine addiction.

## Figures and Tables

**Figure 1 ijerph-19-08602-f001:**
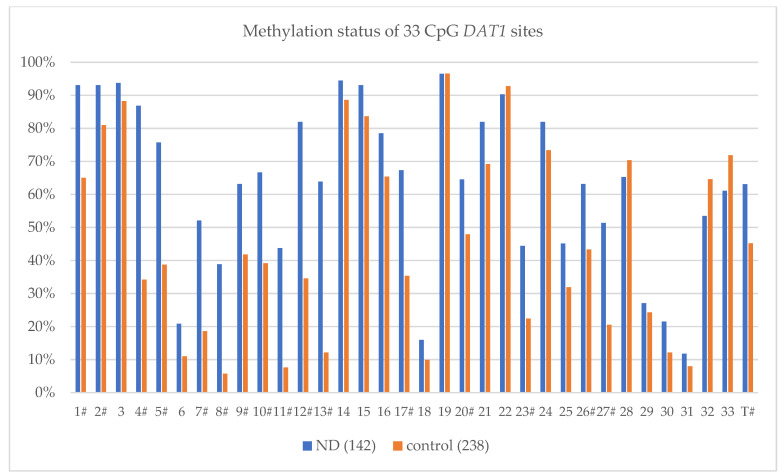
The methylation level (%) of 33 CpG DAT1 sites in a group of nicotine-dependent subjects (ND) and controls. # statistically significant differences in the level of methylation-Bonferroni correction was used, and the *p*-value was reduced to 0.0015 (*p* = 0.05/33 (number of statistical tests conducted)). T—% Total methylation sites.

**Figure 2 ijerph-19-08602-f002:**
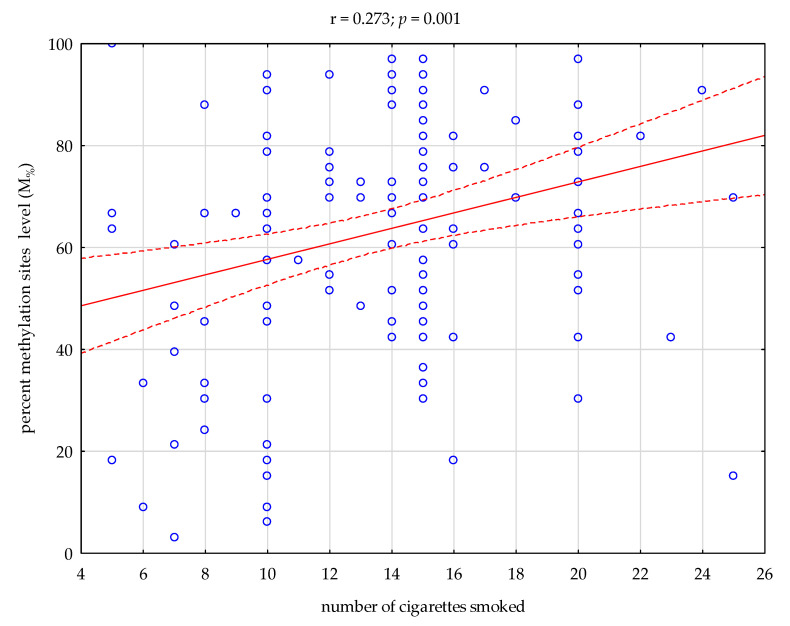
Correlation between percent methylation sites (M_%_) of 33 CpG DAT1 sites and the number of cigarettes smoked in a group of nicotine-dependent subjects (ND).

**Table 1 ijerph-19-08602-t001:** The age of participants in the nicotine-dependent (ND) subjects group and control group.

	**ND**	**Control**	**Mann–Whitney U-Test *(p)***
*n*	142	238	
Age M (SD)	27.57 (10.01)	21.86 (3.55)	−1.726 (0.08432)

*p*-statistical significance with the Mann–Whitney U-test; *n*—number of subjects; M ± SD—mean ± standard deviation.

**Table 2 ijerph-19-08602-t002:** PCR reaction conditions for the amplification of a 447 bp fragment encompassing 33 CpG sites in the promoter of the *DAT1* gene.

Number of Cycles	PCR Step	Temperature	Time
1	Initial denaturation	94 °C	5:00
35	Denaturation	94 °C	0:25
	Annealing	61 °C	0:25
	Elongation	72 °C	0:25
1	Final elongation	72 °C	5:00

**Table 3 ijerph-19-08602-t003:** The methylation status of 33 CpG *DAT1* sites in a group of nicotine-dependent subjects (ND) and controls.

CpG Site	Studied Group	Methylation Level (%)	χ^2^(*p*)	OR	95% CI (−95%, +95%)
**1 #**	ND (142)	93.06%	38.945 (0.000001)	7.209	(3.613, 14.383)
	control (238)	65.02%			
**2 #**	ND (142)	93.06%	10.780 (0.00103)	3.145	(1.542, 6.414)
	control (238)	80.99%			
**3**	ND (142)	93.75%	3.219 (0.07278)	2.004	(0.926, 4.337)
	control (238)	88.21%			
**4 #**	ND (142)	86.81%	103.251 (0.000001)	12.646	(7.327, 21.825)
	control (238)	34.22%			
**5 #**	ND (142)	75.69%	50.779 (0.000001)	4.915	(3.120, 7.744)
	control (238)	38.78%			
**6**	ND (142)	20.83%	7.220 (0.00721)	2.123	(1.216, 3.707)
	control (238)	11.03%			
**7 #**	ND (142)	52.08%	49.153 (0.000001)	4.747	(3.024, 7.451)
	control (238)	18.63%			
**8 #**	ND (142)	38.89%	70.829 (0.000001)	10.479	(5.639, 19.471)
	control (238)	5.73%			
**9 #**	ND (142)	63.19%	16.999 (0.00004)	2.388	(1.572, 3.627)
	control (238)	41.83%			
**10 #**	ND (142)	66.67%	28.168 (0.000001)	3.107	(2.029, 4.756)
	control (238)	39.16%			
**11 #**	ND (142)	43.75%	74.885 (0.000001)	9.450	(5.385, 16.583)
	control (238)	7.60%			
**12 #**	ND (142)	81.94%	83.488 (0.000001)	8.578	(5.229, 14.070)
	control (238)	34.60%			
**13 #**	ND (142)	63.89%	117.503 (0.000001)	12.772	(7.728, 21.105)
	control (238)	12.17%			
**14**	ND (142)	94.44%	3.764 (0.05238)	2.188	(0.975, 4.911)
	control (238)	88.59%			
**15**	ND (142)	93.06%	7.267 (0.00702)	2.619	(1.273, 5.385)
	control (238)	83.65%			
**16**	ND (142)	78.47%	7.576 (0.00591)	1.928	(1.203, 3.091)
	control (238)	65.40%			
**17 #**	ND (142)	67.36%	38.282 (0.00000)	3.772	(2.453, 5.801)
	control (238)	35.36%			
**18**	ND (142)	15.97%	3.255 (0.07121)	1.733	(0.949, 3.164)
	control (238)	9.89%			
**19**	ND (142)	96.53%	0.0007 (0.97882)	0.985	(0.323, 2.996)
	control (238)	96.58%			
**20 #**	ND (142)	64.58%	10.409 (0.00125)	1.982	(1.304, 3.013)
	control (238)	47.91%			
**21**	ND (142)	81.94%	7.797 (0.00523)	2.019	(1.226, 3.326)
	control (238)	69.20%			
**22**	ND (142)	90.28%	0.779 (0.37737)	0.723	(0.351, 1.489)
	control (238)	92.78%			
**23 #**	ND (142)	44.44%	21.378 (0.00000)	2.766	(1.785, 4.287)
	control (238)	22.43%			
**24**	ND (142)	81.94%	3.783 (0.05177)	1.646	(0.993, 2.727)
	control (238)	73.38%			
**25**	ND (142)	45.14%	6.986 (0.00821)	1.753	(1.154, 2.663)
	control (238)	31.94%			
**26 #**	ND (142)	63.19%	14.664 (0.00013)	2.244	(1.478, 3.406)
	control (238)	43.35%			
**27 #**	ND (142)	51.39%	41.095 (0.00000)	4.091	(2.627, 6.372)
	control (238)	20.53%			
**28**	ND (142)	65.28%	1.107 (0.2927)	0.792	(0.513, 1.222)
	control (238)	70.34%			
**29**	ND (142)	27.08%	0.371 (0.54195)	1.154	(0.726, 1.835)
	control (238)	24.33%			
**30**	ND (142)	21.53%	6.231 (0.01255)	1.980	(1.150, 3.407)
	control (238)	12.17%			
**31**	ND (142)	11.81%	1.605 (0.20523)	1.542	(0.786, 3.028)
	control (238)	7.98%			
**32**	ND (142)	53.47%	4.863 (0.02743)	0.628	(0.416, 0.951)
	control (238)	64.64%			
**33**	ND (142)	61.11%	4.948 (0.02611)	0.615	(0.400, 0.945)
	control (238)	71.86%			
**Total Methylation Sites M_T_ ± SD; (M_T%_ ± SD)**					
	ND (142)	20.82 ± 8.01; (63.09% ± 24.28%)			
	control (238)	14.92 ± 5.81; (45.20% ± 17.61%)			
**Mann–Whitney U-Test Z (*p*)**		7.881 (0.000001) #			

χ^2^ (*p*)—chi-square test (significance level); OR—odds ratio; CI—confidence interval (−95%, +95%); R (*p*)—Spearman’s correlation (significance level); Mann–Whitney U-test; *n*—number of subjects; M (SD)—mean (standard deviation); # statistically significant differences in the level of methylation - Bonferroni correction was used, and the *p*-value was reduced to 0.0015 (*p* = 0.05/33 (number of statistical tests conducted)); M_T_—mean total methylation sites; and M_T%_—mean percent total methylation sites.

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
