# Peer review of "Methylation in the Promoter Region of the Dopamine Transporter DAT1 Gene in People Addicted to Nicotine"

_ijerph, 2022, doi:10.3390/ijerph19148602_

Round 1

Reviewer 1 Report

In this study, the authors investigated the methylation of the promoter region of the dopamine transporter DAT1 gene in human samples. Methylation-specific PCR revealed that DAT1 methylation is significantly increased in the smoking group compared with the age-matched control group. This study is a simple experimental design, and the data may provide us with new insights into the biological aspect of nicotine addiction. For publication, however, some concerns need to be addressed. 

In Abstract lines 24-25, “We investigated 141 nicotine-dependent…” is 142?

In Figure1, images are of poor quality, and (A) seems missing.

About the results of Table3, it is better to show the summarized figure of the hot point of methylation site of DAT1promoter. 

Author Response

Dear Reviewer,

Thank you very much for the review and for showing the shortcomings in our Manuscript. We analyzed comments and replied to each, indicating where and what corrections were made to the Manuscript, indicating the line and page.

Below are the point-by-point answers.

With respect

Author's

In this study, the authors investigated the methylation of the promoter region of the dopamine transporter DAT1 gene in human samples. Methylation-specific PCR revealed that DAT1 methylation was significantly increased in the smoking group compared to the control group of the same age. This study is a simple experimental project, and the data may provide us with new insight into the biological aspect of nicotine addiction. However, some concerns need to be addressed when it comes to publishing.

  1. In Abstract lines 24-25, “We investigated 141 nicotine-dependent…” is 142?

 - We apologize for this editorial error. We investigated 142 nicotine-dependent subjects and 238 controls.

  1. In Figure1, images are of poor quality, and (A) seems missing.

 - The figure has been removed; instead of this figure, we have inserted the graph.

  1. About the results of Table3, it is better to show the summarized figure of the hot point of methylation site of DATpromoter.

 - Table 3 has been changed to the appropriate figure.

Reviewer 2 Report

Introduction

Please, write the aim of this paper.

Participants

Epigenetic mechanisms in DAT1 have been shown to contribute to several psychiatric disorders (Attention-deficit/hyperactivity disorder, schizophrenia, Parkinson's disease, and others), Authors considered this an exclusion criterion?

Can you include of cigarette smoking subjects, the following data, cigarettes consumed per day, and Fagerström Test for Nicotine Dependence?

Lines 149-151 “Methylation of cytosine was considered positive when the 150 G/A+G ratio accounted for at least 20% of a total signal”. Can you add references and indicate the reason to utilize the G/A+G ratio accounted for at least 20%?

Please, write the mathematical equation to calculate the percentage of methylation in each participant. Remember including references

Figure 1 was present previously (PMID: 32586035 and 34168698). Can you delete this figure and add the references indicated?

Results of methylation level should be in a repository public. Can you include the link, please?

Lines 202-204. Cigarette consumption is important in the study population because methylation level depends on the degree of exposition to risk factors, can you discuss about this topic?

It would be interesting to carry out a correlation test between the cigarettes consumed per day and the level of methylation.

Can you add in the discussion section about the consequences of CpG island in DAT1 hyper- or hypo-methylated?

Line 211. Can you indicate the meaning of “GR”?

In some lines exist mistakes typing, for example, 187 “homeostasis” or line 206 “canna-bis”

Author Response

Dear Reviewer,

Thank you very much for the review and valuable comments, which we apologize for and for showing shortcomings and nonsense in our Manuscript. We analyzed comments and replied to each, indicating where and how the corrections in the Manuscript were made, indicating the line and page.

Below are the point-by-point answers.

With respect

Authors

  1. Introduction.

- Thanks for the suggestion. The aim has been added. Lines 103-105, page 3.

  1. Participants.

a. Epigenetic mechanisms in DAT1 have been shown to contribute to several psychiatric disorders (Attention-deficit/hyperactivity disorder, schizophrenia, Parkinson's disease, and others), Authors considered this an exclusion criterion?

- Yes, healthy smokers were recruited for the study, with no accompanying psychiatric illnesses, schizophrenia, Parkinson's and others.

b. Can you include of cigarette smoking subjects, the following data, cigarettes consumed per day, and Fagerström Test for Nicotine Dependence?

- Yes, all the data and descriptions were added to the Manuscript - lines 111-112, page 3.

  1. Lines 149-151 “Methylation of cytosine was considered positive when the 150 G/A+G ratio accounted for at least 20% of a total signal”. Can you add references and indicate the reason to utilize the G/A+G ratio accounted for at least 20%?

- Thank you for this question. Of course, we used literature methodologically and showed a calculation tool in the Manuscript. However, it seems justified to ask the Reviewer for a methodological reference. As described by Parrish et al., 2012 (in: Direct bisulfite sequencing for examination of DNA methylation patterns with gene and nucleotide resolution from brain tissues; Curr Protoc. Neurosci 2012) - “Methylation levels for each CpG site within the DNA amplicon can be quantified by measuring the ratio between peak height values of cytosine (C) and thymine (T), yielding the basic equation for the methylation percentage to be (C / (C + T) * 100). Note that this only applies in cases where the forward primer was used for DNA sequencing. If the reverse primer was used, the guanine (G) and adenine (A) peak heights should be used instead, yielding the equation (G / (G + A) * 100). “In our study, there was situation 2, the reverse primer, and we followed the above formula.

  1. Please, write the mathematical equation to calculate the percentage of methylation in each participant. Remember including references

- Of course, the formula is on (G / (G + A) * 100) lines 155-156, page 4. And the references were added: (Parrish et al., 2012)

Parrish, R. R., Day, J. J., & Lubin, F. D. (2012). Direct bisulfite sequencing for examination of DNA methylation with gene and nucleotide resolution from brain tissues. Current Protocols in Neuroscience, 1(SUPPL.60). https://doi.org/10.1002/0471142301.NS0724S60

  1. Figure 1 was present previously (PMID: 32586035 and 34168698). Can you delete this figure and add the references indicated?

- Yes, it was removed in the Manuscript.

  1. Results of methylation level should be in a repository public. Can you include the link, please?

- Thank you, especially for this suggestion. We have added the results to the repository, link below:

https://ppm.pum.edu.pl/info/researchdata/PUM1897b8b56aa74aa1a7892dc50499e096/

  1. Lines 202-204. Cigarette consumption is important in the study population because methylation level depends on the degree of exposition to risk factors, can you discuss about this topic?

- Thanks for this tip. Information added in the Manuscript. Lines 235-254 Page 8.

  1. It would be interesting to carry out a correlation test between the cigarettes consumed per day and the level of methylation.

- This is a valuable comment. We have shown in the text and Figure 2 .; lines184-186 and 192-194, pages 4 and 5.

  1. Can you add in the discussion section about the consequences of CpG island in DAT1 hyper- or hypo-methylated?

- Thank you for this suggestion - the discussion section line 267-272 has been added. Page 8.

  1. Line 211. Can you indicate the meaning of “GR”?

- Of course, as indicated in the text (glucocorticoid receptor), lines 260-261, page 8.

  1. In some lines exist mistakes typing, for example, 187 “homeostasis” or line 206 “canna-bis”

- Thank you for the tip. Bugs are fixed.

Round 2

Reviewer 2 Report

The authors satisfactorily addressed the comments